# ECHO STATE TRANSFORMER:
# ATTENTION OVER FINITE MEMORIES

## ABSTRACT

While Large Language Models and their underlying Transformer architecture are remarkably efficient, they do not reflect how our brain processes and learns a diversity of cognitive tasks such as language and working memory. Furthermore, sequential data processing with Transformers encounters a fundamental barrier: quadratic complexity growth with sequence length. Motivated by these limitations, our ambition is to create more efficient models that are less reliant on intensive computations. We introduce Echo State Transformers (EST), a hybrid architecture that elegantly resolves this challenge while demonstrating exceptional performance in classification and detection tasks. EST integrates the Transformer attention mechanisms with principles from Reservoir Computing to create a fixed-size window distributed memory system. Drawing inspiration from Echo State Networks, the most prominent instance of the Reservoir Computing paradigm, our approach leverages reservoirs (random recurrent networks) as a lightweight and efficient memory. Our architecture integrates a new module called "Working Memory" based on several reservoirs working in parallel. These reservoirs work as independent working memory units with distinct internal dynamics. A novelty here is that the classical reservoir hyperparameters, controlling the dynamics, are now trained. Thus, the EST dynamically adapts the reservoir memory/non-linearity trade-off. Thanks to these working memory units, EST achieves constant computational complexity at each processing step, effectively breaking the quadratic scaling problem of standard Transformers. We evaluate ESTs on a recent challenging timeseries benchmark: the Time Series Library, which comprises 69 tasks across five categories. Results show that ESTs ranks first overall in two of five categories, outperforming strong state-of-the-art baselines on classification and anomaly detection tasks, while remaining competitive on short-term forecasting. These results position ESTs as a compelling alternative for time-series classification and anomaly detection, and a practical complement to transformer-style models in applications that prioritize robust representations and sensitive event detection.

## 1 INTRODUCTION

Transformers (Vaswani et al., 2017) have constituted a genuine revolution in the field of artificial intelligence, offering for the first time a scalable architecture capable of efficiently processing text while overcoming the inherent constraints of classical Recurrent Neural Networks (RNNs) and their costly training related to backpropagation through time (Werbos, 1990). This major innovation, however, comes with a significant drawback: a quadratic complexity that increases with the length of the sequence to be processed.

Although Transformers break free from RNN formalism and their internal states – that allows information to be transmitted from time $t$ to time $t + 1$ – they still need to access previous information. To achieve this, rather than using only the current information, they mobilize the entire input sequence at each step of the processing, meaning a Transformer has no internal memory per se, but must process the entire sequence at each time step. This characteristic generates several major problems. The most obvious concerns the energy and computational costs that inexorably increase as the conversation with a Transformer lengthens. A second problem lies in the fact that a Transformer, like a human being, eventually becomes overwhelmed when the sequence becomes particularly long

(Modarressi et al., 2025). Moreover, having to retain the entire sequence is equivalent to possessing an "infinite" working memory, which is considerably far from what is biologically plausible (Baddeley, 1992). Indeed, when we humans read a book, we are incapable of memorizing all the words and punctuation elements that compose it. Instead, we have a "finite" memory in which we "compress" information, even if it means sacrificing certain aspects. Biological systems show us that efficient ways to manage memory are possible, thus we take inspiration from Working Memory models (Strock et al., 2020).

In this paper, we propose to add a Working Memory block to the standard Transformer architecture (See Fig. 1). Our method is inspired by both Transformers and Reservoir Computing (Lukoševičius & Jaeger, 2009; Yan et al., 2024) to exploit the attention mechanisms characteristic of Transformers, not on the entire input sequence, but on a finite set of memory units. Each of these units has its own dynamics and must retain a specific part of the information received as input. Thus, since the number of memory units is determined at the initialization of the model and remains constant (unlike the number of tokens in the input sequence which continuously grows), the complexity of the attention mechanism also becomes constant. During each update step, each memory unit creates its Queries from the current input embedding at time $t$, while generating Keys and Values based on both its own state and the states of other memory units. This mechanism allows each unit to independently extract from the collective memory the information it deems relevant and compare it to the input. Once the memories are updated, a second attention mechanism allows their content to be exploited to predict the output at time $t$.

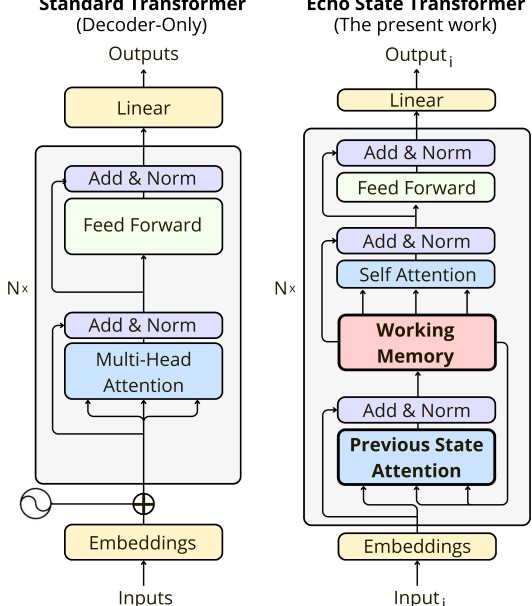

Figure 1: Comparison of standard Transformer (Decoder-Only) and Echo State Transformer architecture. We add a "Working Memory" block to the standard Transformer architecture and apply attention on it with the "Previous State Attention" block. This block computes Keys and Values from the previous state, and Queries from the input at time $t$.

## 2 RELATED WORKS

Numerous approaches have been explored to address the complexity of Transformers. Among them, many seek to modify, replace, or eliminate the calculation of attention, particularly the *softmax* which is the operation requiring the entire sequence to be reprocessed at each time step. Thus, Performers (Choromanski et al., 2020) estimate the attention matrix with linear complexity using FAVOR+. Attention Free Transformers (AFT) (Zhai et al., 2021) eliminate self-attention by first combining Keys and Values with a set of positional biases before multiplying the result with Queries, achieving linear complexity. Reformer (Kitaev et al., 2020) replaces the attention product with locality-sensitive hashing offering linearithmic complexity. RWKV (Peng et al., 2023) replaces attention and MLP blocks with two blocks allowing the mixing of temporal and spatial information, thus offering a linear solution in time and space. Retentive Network (RetNet) (Sun et al., 2023) replaces the softmax of attention with an exponential decay applied to the result of the product of Queries and Keys, allowing three modes of usage including linear complexity inference. Mamba (Gu & Dao, 2023) proposes an innovative architecture based on selective State Space Models (SSMs) and allows linear training and linear inference. TimesNet (Wu et al., 2022) reshapes time series into 2D tensors over learned periods and applies lightweight convolutional inception blocks to capture temporal patterns, which leads to a linear complexity in sequence length.

Other approaches exist that do not seek to modify the attention calculation but rather to redesign how the input sequence is represented. This is the case with TransformerFAM (Hwang et al., 2024)

which introduces a feedback loop designed to dynamically create two tokens representing past information, thus creating an internal memory to process sequences of indefinite length. iTransformer (Liu et al., 2023) inverts tokenization by treating each variate as a token instead of each time step, enabling attention to capture cross-variate dependencies while feed-forward networks learn temporal patterns. PatchTST (Nie et al., 2022) splits each univariate series into patches and applies channel-independent attention, enabling longer lookback windows and improved long-term forecasting. Large Memory Model (LM2) (Kang et al., 2025) proposes a memory module capable of interacting with input tokens and updating itself via gate mechanisms, similar to LSTMs (Hochreiter & Schmidhuber, 1997), to store and compress past information.

## 3 ARCHITECTURE INSPIRATION

Our Echo State Transformer architecture draws inspiration from two powerful paradigms in sequential processing: Transformers and Reservoir Computing. This section briefly introduces the key components from each that we leverage in our hybrid model.

### 3.1 TRANSFORMERS

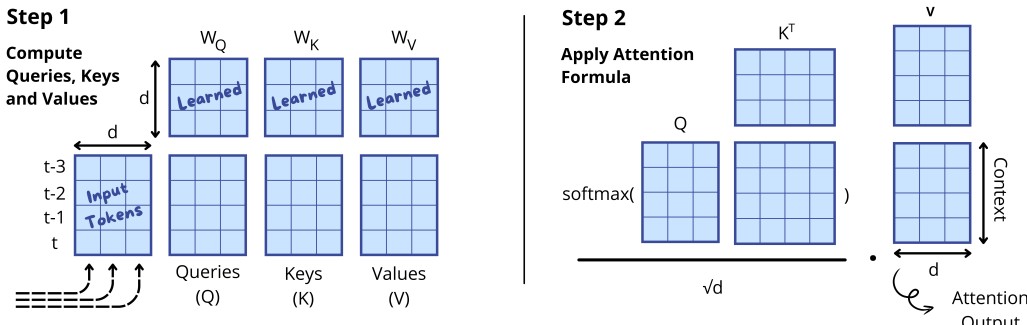

Figure 2: Computation of attention in Transformers exemplified with matrix multiplications schemas. On the left the computation of Queries, Keys and Values. On the right, the application of the attention formula with the previous computed Queries, Keys and Values.

Transformers revolutionized sequence processing by replacing recurrence with attention mechanisms, allowing direct interaction between any positions in a sequence. The core innovation of Transformers is the self-attention mechanism:

$$\text{Attention}(Q, K, V) = \text{softmax}\left(\frac{QK^T}{\sqrt{d_k}}\right) V \tag{1}$$

where $Q$ (queries), $K$ (keys), and $V$ (values) are linear projections of the input. This mechanism computes a weighted sum of values, with weights determined by the compatibility between queries and keys. Transformers process sequences through multiple layers, each containing a Self-Attention and Feed-Forward modules. Recent research suggests these feed-forward layers act as key-value memories (Geva et al., 2020), like one might observe in the brain (Gershman et al., 2025), storing knowledge acquired during training.

While Transformers excel at modeling complex dependencies and can process sequences in parallel, they face a quadratic complexity challenge with sequence length and lack an inherent mechanism for maintaining state across processing steps.

### 3.2 RESERVOIR COMPUTING

Reservoir Computing, particularly Echo State Networks (ESNs) (Jaeger, 2001a; Jaeger & Haas, 2004) offers an efficient approach to sequential processing through fixed random recurrent networks. The key dynamics of a reservoir are captured by:

$$\mathbf{s}_t = (1 - \alpha).\mathbf{s}_{t-1} + \alpha.f(\mathbf{W}_{in}.\mathbf{u}_t + \mathbf{W}.\mathbf{s}_{t-1}) \tag{2}$$

where $\mathbf{s}_t$ is the reservoir state, $\mathbf{u}_t$ is the input, $\mathbf{W}_{in}$ and $\mathbf{W}$ are fixed random matrices, $f$ is a non-linear activation (typically $\mathtt{tanh}$), and $\alpha$ is the leak rate. Two critical parameters govern reservoir behavior: the spectral radius – defined as the largest absolute eigenvalue of $\mathbf{W}$ – controls the Echo State Property (Yildiz et al., 2012), with values below 1 ensuring stability and values approaching 1 maximizing memory capacity while maintaining predictable behavior; complementing this, the leak rate ($\alpha$) regulates how quickly the reservoir state evolves (Jaeger et al., 2007), where lower values preserve previous states longer and higher values enhance the network's responsiveness to new inputs.

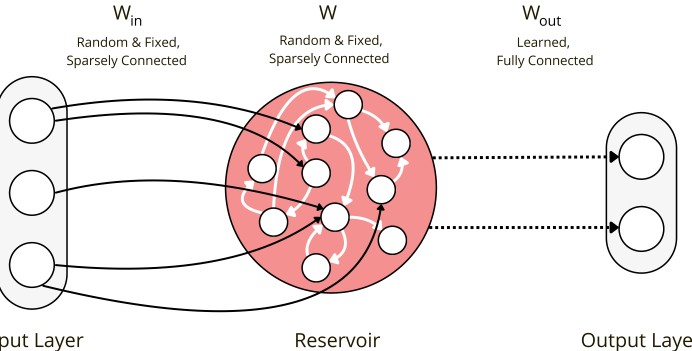

Figure 3: Echo State Network is composed of 3 layers: $W_{in}$ treats the input, $W$ computes the state update and $W_{out}$ computes the output. Only $W_{out}$ is trained via linear regression.

Reservoirs excel at maintaining temporal information with minimal parameter tuning and training data requirements (Lukoševičius & Jaeger, 2009). By only training output weights, ESNs achieve remarkable efficiency while maintaining powerful temporal processing capabilities (Jaeger, 2001a). If properly configured, reservoirs can retain temporal information proportional to their size (Ceni & Gallicchio, 2024). More details can be found in Appendix B.

## 4 MODEL ARCHITECTURE

Our approach, the Echo State Transformer (EST)[1], proposes a hybrid architecture inspired by original Transformers and the Reservoir Computing paradigm. The main objective is to overcome the quadratic complexity inherent to traditional Transformers by introducing a working memory composed of a finite number of units where attention will be applied, instead of applying it to a potentially infinite number of input tokens, thus making the complexity constant in time and space for inference.

The EST model mainly consists of six distinct blocks: an input block, an attention block on the previous state, a working memory block, a self-attention block, a feed-forward block, and an output block. A detailed schema of the architecture is available in the Appendix A.

- **Input**: This block receives the current token at time $t$. This token is then transformed into an embedding, preparing the information for the subsequent layers.

- **Previous State Attention**: This block allows each unit of the Working Memory to create a unique information vector intended for it (see Fig. 4). To do this, each unit creates a query from the embedding coming from the Input Block as well as keys and values from the state of all memory units, then calculates the attention by adding the residuals (embedding). The underlying idea is that each memory unit can access its own state as well as that of other units to identify relevant information in the current input and supplement its knowledge. The attention mechanism allows weighting the dimensions of the input embedding according to the relevance of the information to adapt it to each memory unit.

- **Working Memory**: The working memory (Fig. 5) comprises multiple configurable reservoir units derived from Reservoir Computing. Each reservoir — a recurrent network with fixed random weights — functions as a dynamic system retaining temporal information

---

[1] Public git repository : https://anonymous.4open.science/r/EchoStateTransformer/

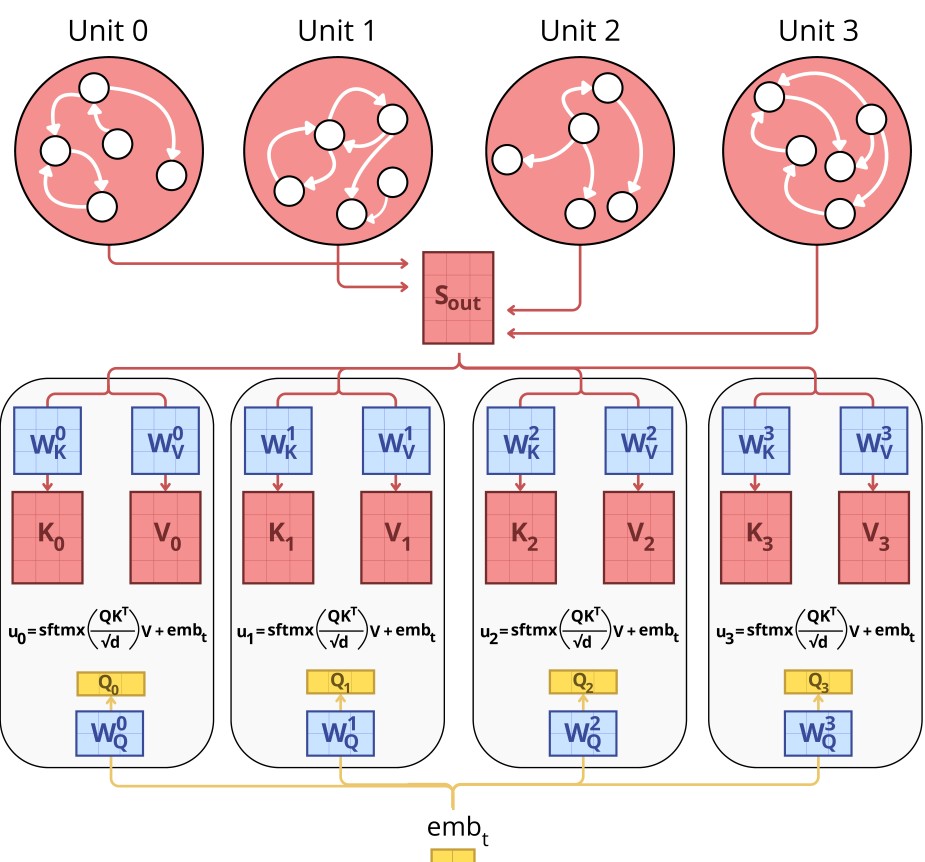

Figure 4: This figure display the mecanism of the Previous State Attention block. It produces Keys and Values from all memory units ($S_{out}$) and Queries from the embedding at time $t$ ($emb_t$). Similarly to Transformer and its Multi-Head Attention block, we compute several distinct products of attention – one per memory unit – that allows each unit to compute its own input vector.

through an echo mechanism. Uniquely, each unit learns its own dynamics parameters (including its spectral radius) through backpropagation and composes information vectors from current inputs and states of all units.

Our approach introduces an adaptive leak rate mechanism that determines how much information persists between time steps (0 preserves the entire previous state, while values approaching 1 favor new information). This adaptivity uses a softmax function: scores calculated from each unit's information vector are normalized to produce weights between 0 and 1, modulating individual leak rates. This competitive mechanism potentially allows certain units to maintain fixed information over time by assigning near-zero leak rates.

- **Self-Attention**: Once the memories are updated, a self-attention layer is applied to all units of the working memory. Each state vector from the memory units is treated as an input token would be in Transformers, for which we compute queries, keys, and values. Attention is then applied and, like Transformers, residual connections from the current state are added to the result of the self-attention. A linear combination is then applied to the result to reduce the dimensions (see Appendix A).

- **Feed Forward (FF)**: Like Transformers, this layer allows the integration of knowledge learned during training (Geva et al., 2020). It is typically implemented using feed-forward layers that increase their dimension (for example, by a factor of 4) before reducing it to the initial size (see Appendix A).

- **Output Layer**: The last layer applies a transformation (potentially the inverse of the embedding matrix) to the output of the Feed Forward to reconstruct the final output.

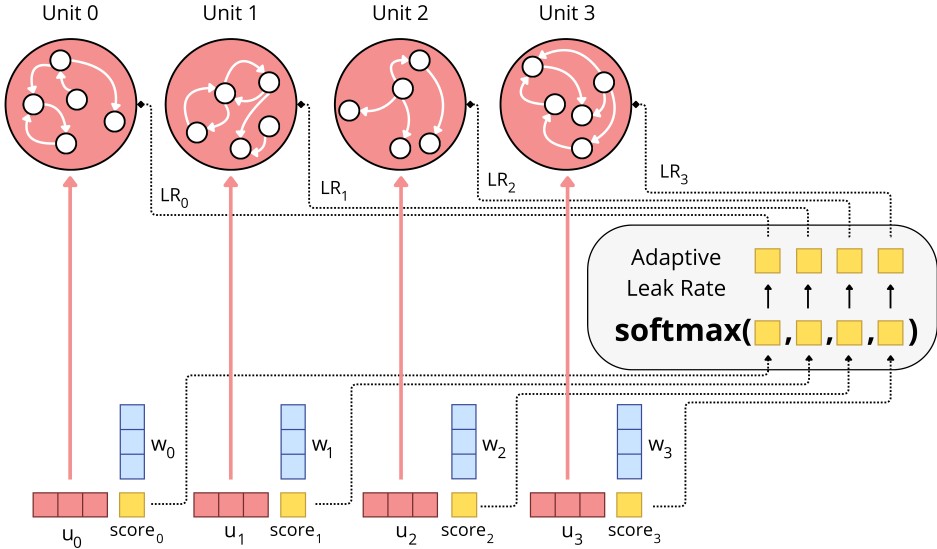

Figure 5: This figure displays the mechanism behind the Working Memory block and more particularly the adaptive leak rate. Each memory unit compute a score from its input vector. Then a softmax is applied on all of this score to compute the leak rate for each unit.

Like Transformers, the previous state attention, working memory, self-attention, and feed-forward blocks can be stacked sequentially to form layers that can be repeated multiple times. However, unlike Transformers, which allow complete parallelization of the entire sequence, ESTs require sequential training like GRUs and LSTMs and thus, backpropagation through time.

The main innovation of this work lies in the introduction of a working memory system where multiple units, based on the Reservoir Computing paradigm, work in parallel to preserve temporal information. Furthermore, the use of an adaptive leak rate allows better control of information retention over time, compensating for the gradual dissipation of the echo. By allowing each memory unit to consult the state of others, the model acquires an increased ability to process information taking into account the global context of the memory. In preliminary works, we tested different architectures that are not included in this paper[2]. We observed that each memory unit tends to stabilize around a characteristic leak rate value, oscillating dynamically in response to input variations. This emergent property allows each unit to maintain a consistent temporal profile while still adapting to contextual needs. Although we tried other non-linear functions (like sigmoid) to replace the softmax and remove the competition between units in the adaptive leak rate, the model did not converge with them.

## 5 EXPERIMENTS

### 5.1 BENCHMARK FRAMEWORK

We evaluate our model on the Time Series Library (TSL) benchmark (Wang et al., 2024), which provides a unified evaluation protocol for deep time series models. TSL covers five task categories, each comprising multiple tasks, for a total of 69 tasks (all tasks are described in Appendix C):

- **Anomaly Detection** (5 tasks): assesses sensitivity to rare and critical deviations in temporal dynamics.

- **Classification** (10 tasks): tests the discriminative power of the learned representations.

- **Imputation** (12 tasks): measures robustness in reconstructing missing data points.

---

[2]To remain anonymous, our previous work will not be cited here.

- **Long-term Forecasting** (36 tasks): evaluates the ability to capture long term temporal dependencies and predict unseen future values.
- **Short-term Forecasting** (6 tasks): evaluates the ability to capture short term temporal dependencies and predict unseen future values.

For each category, TSL specifies standard metrics: Mean Squared Error (MSE) for long-term forecasting and imputation, Overall Weighted Average (OWA) for short-term forecasting, Accuracy for classification, and F1-score for anomaly detection. This large-scale evaluation setting enables a comprehensive comparison across diverse models and tasks.

## 5.2 EVALUATION SETUP

We compare the proposed Echo State Transformer (EST) with state-of-the-art baselines present in Wang et al. (2024) including iTransformer, PatchTST, FEDformer, TimesNet, Mamba, Reformer, DLinear, and N-BEATS, among others. All experiments strictly follow the training protocols defined in TSL to ensure fair comparison, including optimization strategies, batch construction, and evaluation methodology. Following the TSL methodology, we tested ten different configurations (e.g. number of layers, memory units, neurons, connectivity) and retained the best-performing one for each task. Since EST belongs to the family of Recurrent Neural Networks (RNN), training requires Backpropagation Through Time (BPTT), unlike Transformer-based approaches which rely on full sequence parallelization. This BPTT implies that every gradients among the whole sequence have to be retained in memory during training (even for the weights that are not learned) leading to a high memory footprint (see Appendix F).

## 6 RESULTS

We evaluate EST across 69 tasks grouped in 5 time series problem categories, comparing against strong baselines including state-of-the-art models. Figure 6 summarizes the comparative performance, revealing EST's distinctive strengths and architectural insights. EST ranks $1^{st}$ overall in 2 of the 5 task categories, demonstrating competitive performance across diverse scenarios. For each task, the optimal configuration (selected among 10) and its performance score are fully reported in Appendix E.

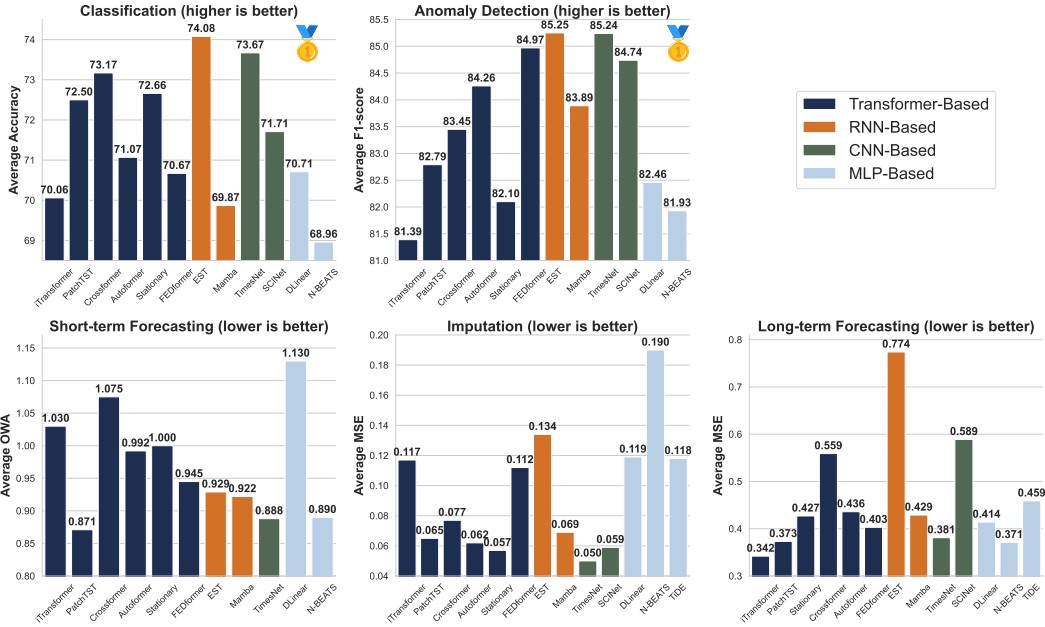

Figure 6: EST vs. baselines across 5 task categories (TSL)

## 6.1 EST ACHIEVES SOTA PERFORMANCE ON CLASSIFICATION AND ANOMALY DETECTION

EST seems to excel at identifying minor variations within complex data streams: it is more sensitive to weak signals and rare events than other methods. In classification, EST achieves 74.08% average accuracy, ranking $1^{st}$ overall among all evaluated models and outperforming leading approaches including TimesNet (73.67%), Crossformer (73.17%), and PatchTST (72.50%). Similarly for anomaly detection, EST secures the top position with an F1-score of 85.25%, maintaining a tight lead over TimesNet (85.24%) and surpassing FEDformer (84.97%). These results demonstrate EST's superior capability in temporal pattern analysis, excelling both at categorical classification and at detecting deviations from normal behavior.

## 6.2 COMPETITIVE PERFORMANCE ON SHORT-RANGE FORECASTING

In short-term forecasting, EST achieves an OWA score of 0.929, ranking 5th among evaluated models. While not leading this category, EST performs competitively with established methods like Mamba (0.922) and substantially outperforms several baselines including iTransformer (1.030) and Crossformer (1.075). This suggests EST effectively captures local temporal dependencies over modest prediction horizons.

## 6.3 ARCHITECTURE INSIGHTS FROM OPTIMAL CONFIGURATIONS

Analysis of the best-performing configurations reveals interesting architectural patterns across task types. For classification and anomaly detection tasks EST performs best with deeper architectures of 2-4 layers, suggesting that hierarchical feature extraction benefits pattern recognition. In contrast, imputation and short-term forecasting predominantly select shallow 1-layer configurations, indicating these tasks benefit from very short-term temporal modeling. Additionally, as expected, certain tasks require greater memory capacity than others. Short-term forecasting tasks consistently favor balanced configurations where memory dimensions closely match attention computation dimensions (e.g., mem64-dim64, mem32-dim32), suggesting that short-term information retention benefits from symmetric representational capacity. In contrast, tasks requiring long-term information retention exhibit a clear preference for substantially larger memory dimensions relative to model dimensions (e.g., mem128-dim64, mem512-dim128). This pattern is consistent with the expectation that larger reservoir enable longer information retention, providing the extended temporal context necessary for these tasks. More broadly, optimal ESTs seems to be usable in various scales, with a minimum of 4 memory units up to 16 units depending on task complexity. Configurations with only 2 memory units are effective only in rare cases, typically for simple patterns or when coupled with high-dimensional memory spaces (e.g., 512mem). These configuration patterns demonstrate EST's architectural versatility: the model's memory profile can be adapted from short-term focused (balanced dimensions) to long-term specialized (large memory dimensions), while simultaneously having its feature extraction complexity adjusted through layer depth and model dimensions. This flexibility allows EST to adapt to different task requirements, leveraging the salient information retention capabilities inherited from Reservoir Computing and the contextual representation learning derived from Transformer attention mechanisms.

## 6.4 LIMITATIONS ON RECONSTRUCTION TASKS

EST shows weaker performance on tasks requiring precise value reconstruction. In imputation, EST achieves an MSE of 0.134, ranking among the least effective compared to specialized models like TimesNet (0.050), SCINet (0.059), and Stationary (0.057). For long-term forecasting, EST's MSE of 0.774 significantly underperforms leading Transformer models such as iTransformer (0.342) and PatchTST (0.373). This reflects EST's architectural priorities: EST's reservoir-based dynamics and adaptive memory harness the expressivity of high-dimensional random feature mappings to capture salient patterns (Cuchiero et al., 2022), deliberately trading exact pointwise reconstruction for richer representational power. Reservoirs are known to successfully capture the underlying attractor of a dynamical system and reproduce its climate, the long-term statistical structure of the dynamics (Lu et al., 2018), but not necessarily the exact pointwise reconstruction. The reliance on BPTT for long sequences may also limit EST's ability to model complex long-range dependencies required for extended horizon forecasting.

# 7    FLOPs COMPARISON

We compute theoretical FLOPs per forward pass by summing primitive operations across shared components (input/temporal embeddings, normalization, linear projections, feed-forward layers, activations, softmax) and model-specific blocks (e.g., attention pattern, SSM updates, convolutions). FLOPs count all additions and multiplications from embedding/normalization, linear projections (QKV, FF, heads), activations (GELU, tanh) and softmax.    We exclude dropout, reshapes/permutes/concats. To focus on scaling in sequence length, all models in Fig. 7 use similar depth/width (e.g.  number of layers, neurons), resulting in ≈1M parameters.

Figure 7: Theoretical FLOPs vs. sequence length for models normalized to $\approx$ 1M parameters.

The computational complexity analysis reveals distinct scaling behaviors across different architectures as sequence length $\mathcal{O}(L)$ increases. Transformer and PatchTST exhibit quadratic scaling $\mathcal{O}(L^2)$ due to their self-attention mechanism operating over $L$ tokens. This quadratic growth dominates computational costs at longer sequences. In contrast, several architectures achieve more favorable scaling properties. Mamba maintains linear complexity $\mathcal{O}(L)$, with its curve positioned slightly above Reformer. Reformer's theoretical complexity is $\mathcal{O}(L \log L)$ but in practice the logarithmic term contributes to relatively few operations compared to the linear term, making it more linear. TimesNet and iTransformer both exhibit hybrid scaling: TimesNet's FFT operations dominate below $\approx 100$ tokens before lightweight 2D convolutions yield linear growth, while iTransformer has nearly constant computational cost ($\approx$ 0.11 GFLOPs) for sequences up to $\approx 500$ tokens, after which linear scaling emerges due to $\mathcal{O}(N^2)$ complexity in the number $N$ of variables (fixed) instead of $\mathcal{O}(L^2)$ in sequence length. Our proposed EST model similarly achieves $\mathcal{O}(L)$ complexity by attending over a fixed number of memory units instead of time tokens. The curve closely matches other linear methods, particularly Mamba. This linear scaling property enables EST to handle long sequences efficiently while preserving the representational benefits of attention-based architectures.

# 8    CONCLUSION

We introduced the Echo State Transformer (EST), a recurrent architecture that combines attention over a finite set of working memory units with reservoir dynamics and an adaptive leak rate mechanism. By attending to a fixed set of memory units rather than a growing token sequence, EST attains linear complexity and maintains a persistent working state that retains the most salient information from the sequence due to reservoir's sensitivity on rare events. On the large TSL benchmark (69 tasks across five categories), EST displays a clear task-dependent profile: it achieves state-of-the-art performance in both classification and anomaly detection, ranking 1st overall in both categories, while remaining competitive for short-term forecasting (5th overall). In contrast, it lags on long-term forecasting and imputation, which demand long-horizon extrapolation and precise pointwise reconstruction. Overall, (1) the results position EST as a leading architecture for time-series classification and anomaly detection tasks – its adaptive working memory enables robust temporal representations, making it well-suited for monitoring and detection tasks requiring sensitivity to rare deviations –, and (2) the results confirm that limiting attention to a fixed set of memory units offers a practical and powerful solution to the quadratic complexity of the QKV attention used in Transformer models. Future work will focus on removing BPTT for full sequence parallelization, requiring the removal of recurrent connection and the linearization of reservoir, potentially preserving discriminative performance while reducing memory costs and accelerating training.

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

REPRODUCIBILITY

To ensure full reproducibility, we provide the complete codebase on an anonymous GitHub repository: https://anonymous.4open.science/r/EchoStateTransformer/. This archive contains the EST model implementation, TSL benchmark integration, computational complexity analysis tools (FLOPs computation), and all visualization scripts used to generate the figures in this paper. The comprehensive set of hyperparameter configurations tested across all runs is documented both within the GitHub repository and in Appendix D.

ETHICS STATEMENT

This work introduces a novel neural architecture, the Echo State Transformer (EST), and evaluates it exclusively on publicly available datasets from the Time Series Library benchmark. No private, sensitive, or personally identifiable data were used in this study. The potential societal risks primarily relate to downstream applications of time-series modeling, such as in healthcare, finance, or critical infrastructure. While EST demonstrates strong performance in classification and anomaly detection, there remains a risk of amplifying biases or producing incorrect alerts if deployed without careful validation. We encourage practitioners to critically evaluate model outputs, consider fairness and bias mitigation strategies, and assess ethical implications before deployment in critical contexts.

LLM USAGE

We utilized large language models to assist with writing this paper: translation from our native language to English, phrasing suggestions, and code completion tasks.

# A  DETAILED SCHEMA OF THE ARCHITECTURE ECHO STATE TRANSFORMER

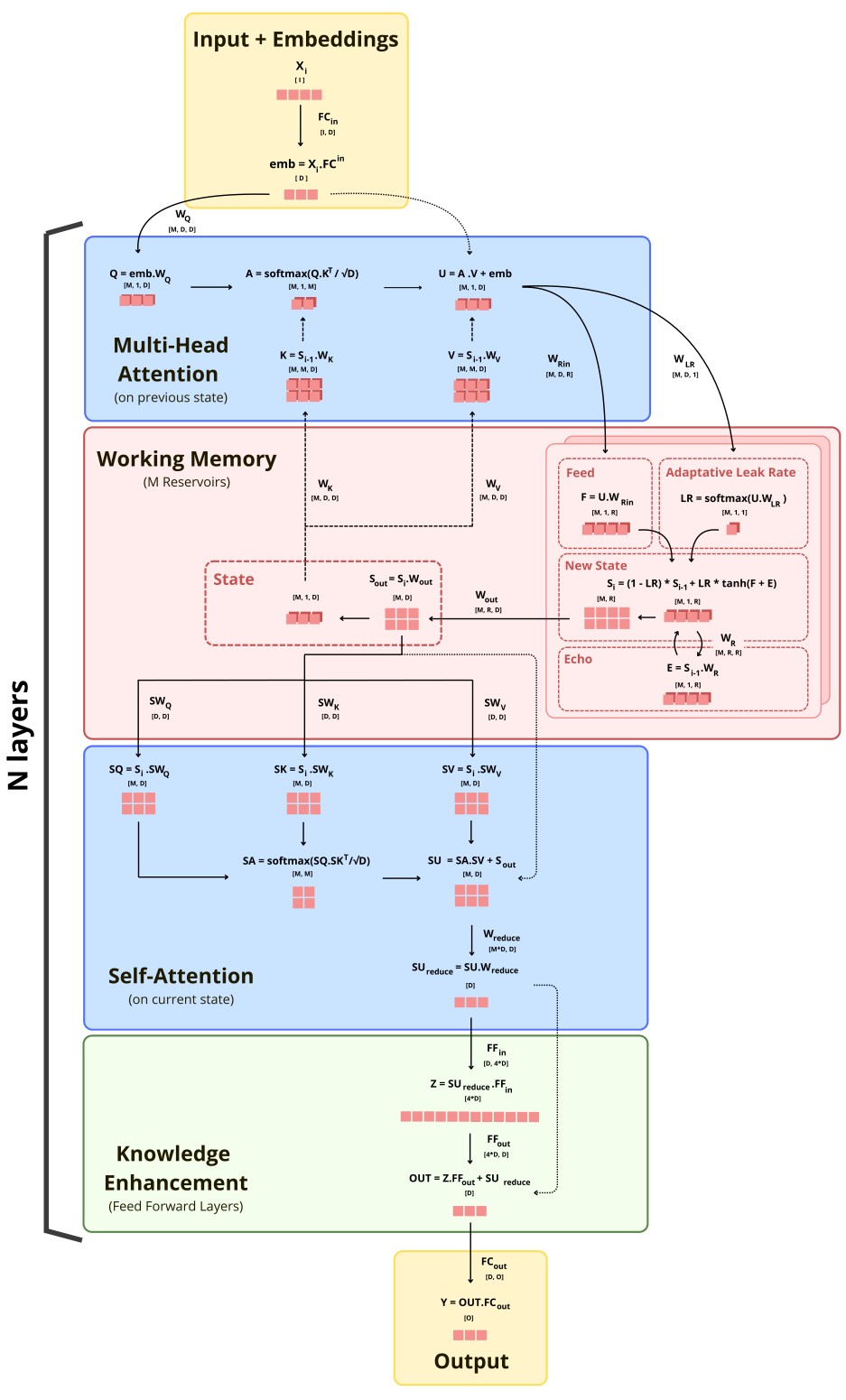

Figure 8: Detailed architecture of Echo State Transformer

# B  Reservoir Computing

**Reservoir Computing (RC)** (Nakajima & Fischer, 2021) is a computing paradigm inspired by recurrent neural networks (RNN) that considerably simplifies the learning process. The main idea is to use a large recurrent neural network, called a "*reservoir*", whose internal connections are fixed and random. Only the connections between the reservoir and the output layer are trained.

## B.1  Why Reservoir Computing?

Reservoir Computing, and specifically Echo State Networks (ESNs), offers several compelling advantages that address the limitations of standard sequential models. ESNs excel at encoding and retaining information over extended temporal sequences through their unique internal dynamics, where recurrent connections within the reservoir maintain a persistent trace of input history (Jaeger, 2001b). This temporal memory capacity scales remarkably well with reservoir size — certain configurations can preserve temporal information proportional to the number of reservoir neurons (Ceni & Gallicchio, 2024). The computational efficiency of ESNs represents another significant advantage. By limiting training to only the output layer while keeping reservoir connections fixed, ESNs dramatically reduce both computational requirements and the volume of training data needed compared to fully-learned recurrent neural networks (Lukoševičius & Jaeger, 2009). This reduced training complexity makes ESNs particularly valuable in resource-constrained environments.

Furthermore, the random matrices at the core of reservoir computing create rich, non-linear dynamic systems that effectively project inputs into high-dimensional spaces where complex patterns become more linearly separable. Our Echo State Transformers model leverages these strengths by implementing a distributed *Working Memory* composed of multiple reservoir modules, combining the efficient temporal processing capabilities of Reservoir Computing with the attention mechanisms that have made Transformers so effective.

### B.1.1  Spectral radius: guardian of the Echo State Property

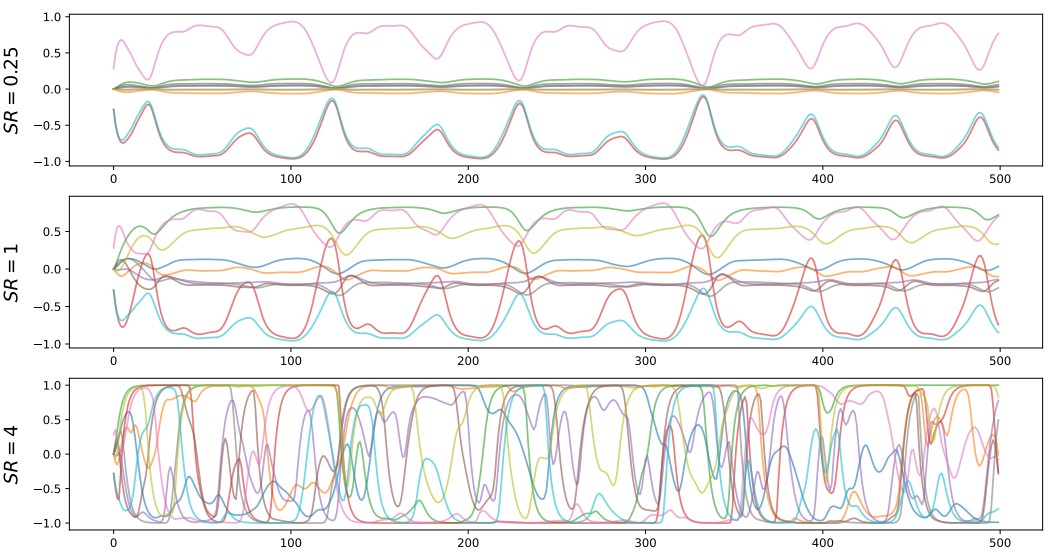

Figure 9: Effect of Spectral Radius on reservoir activity.

The spectral radius, defined as the largest absolute eigenvalue of the reservoir matrix, is critical for maintaining the Echo State Property (Jaeger, 2001a) (Yildiz et al., 2012). When below 1, this property ensures initial conditions' influence gradually fades. The spectral radius creates a fundamental trade-off (see Fig. 9): values approaching 1 maximize memory capacity for longer-term dependencies, while lower values enhance stability at the cost of memory. Exceeding 1 progressively leads to saturation of the reservoir state and pushes the network into chaotic regimes where minimal input differences produce dramatically different outputs — the "butterfly effect" — rendering the

system unreliable for sequential processing. Optimal performance typically occurs at the "edge of chaos" (spectral radius just below 1) where computational capacity is maximized without sacrificing stability.

### B.1.2 LEAK RATE: CONTROL OF TEMPORAL DYNAMICS

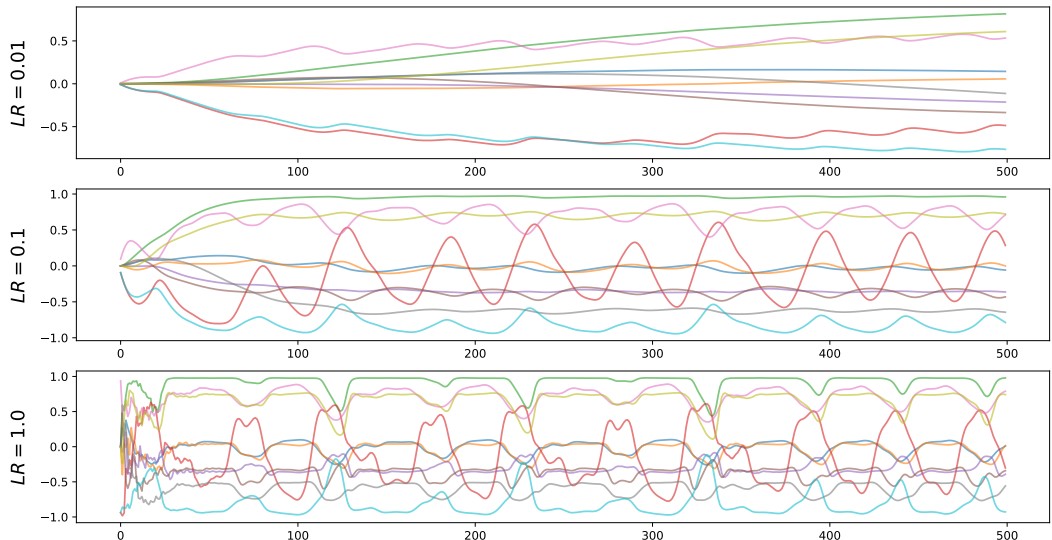

Figure 10: Effect of Leak Rate on reservoir activity.

The leak rate $\alpha \in (0, 1]$ allows control of the speed at which the reservoir state forgets past information (Jaeger et al., 2007), acting as a low-pass filter on the temporal evolution of states (see Fig. 10). A low value of $\alpha$ (close to 0) slows down the network dynamics, allowing it to process slowly evolving signals or capture longer-term dependencies: the previous state is almost entirely preserved. Conversely, a high value of $\alpha$ (close to 1) accelerates the dynamics, making the network more responsive to rapid changes in the input signal: the previous state is almost entirely replaced by the new state.

### B.2 ECHO STATE NETWORK (ESN)

The most popular model of Reservoir Computing is the Echo State Network (ESN) (Jaeger, 2007) (see Fig. 3). An ESN consists of three distinct parts:

- **Input Layer**: A random, fixed and sparse matrix that projects input data into the reservoir.
- **Reservoir**: A random, fixed and sparse matrix that interconnects neurons recurrently.
- **Output Layer**: A learned matrix that combines the states of the reservoir to produce the desired output. The weights of this matrix are the only part of the network that is trained.

## C   TIME SERIES LIBRARY (TSL BENCHMARK)

All tasks per category are described in the following Table 1, this table is extracted from Wang et al. (2024).

Table 1: Datasets used in Time Series Library

| Task | Dataset | Dimension | Length | Domain | Size |
|------|---------|-----------|--------|--------|------|
| **Classification** | EthanolConcentration | 3 | 1,751 | Alcohol Industry | 20.3 MB |
| | FaceDetection | 144 | 62 | Face (250 Hz) | 789.1 MB |
| | Handwriting | 3 | 152 | Motion | 3.9 MB |
| | Heartbeat | 61 | 405 | Health (0.061s) | 87.8 MB |
| | JapaneseVowels | 12 | 29 | Voice | 1.1 MB |
| | PEMS-SF | 963 | 144 | Transportation (1 day) | 420.1 MB |
| | SelfRegulationSCP1 | 6 | 896 | Health (256 Hz) | 17.8 MB |
| | SelfRegulationSCP2 | 7 | 1,152 | Health (256 Hz) | 17.7 MB |
| | SpokenArabicDigits | 13 | 93 | Voice (11025 Hz) | 37.6 MB |
| | UWaveGestureLibrary | 3 | 315 | Gesture | 3.4 MB |
| **Imputation** | ETTh1, ETTh2 | 7 | 17,420 | Electricity (1 hour) | 10.4 MB |
| | ETTm1, ETTm2 | 7 | 69,680 | Electricity (15 mins) | 2.6 MB |
| | Electricity | 321 | 26,304 | Electricity (1 hour) | 95.6 MB |
| | Weather | 21 | 52,696 | Environment (10 mins) | 7.2 MB |
| **Long-term Forecasting**[3] | ETTh1, ETTh2 | 7 | 17,420 | Electricity (1 hour) | 10.4 MB |
| | ETTm1, ETTm2 | 7 | 69,680 | Electricity (15 mins) | 2.6 MB |
| | Electricity | 321 | 26,304 | Electricity (1 hour) | 95.6 MB |
| | Weather | 21 | 52,696 | Environment (10 mins) | 7.2 MB |
| | Traffic | 862 | 17,544 | Transportation (1 hour) | 136.5 MB |
| | Exchange | 8 | 7,588 | Economic (1 day) | 623 KB |
| | ILI | 7 | 966 | Health (1 week) | 66 KB |
| **Short-term Forecasting** | M4-Yearly | 1 | 6 | Demographic | 589.5 MB |
| | M4-Quarterly | 1 | 8 | Finance | 589.5 MB |
| | M4-Monthly | 1 | 18 | Industry | 589.5 MB |
| | M4-Weekly | 1 | 13 | Macro | 589.5 MB |
| | M4-Daily | 1 | 14 | Micro | 589.5 MB |
| | M4-Hourly | 1 | 48 | Other | 589.5 MB |
| **Anomaly Detection** | SMD | 38 | 100 | Industry (1 min) | 436.4 MB |
| | MSL | 55 | 100 | Industry (1 min) | 58.2 MB |
| | SMAP | 25 | 100 | Industry (1 min) | 113.0 MB |
| | SwaT | 51 | 100 | Industry (1 min) | 903.2 MB |
| | PSM | 25 | 100 | Industry (1 min) | 107.1 MB |

**Anomaly Detection (5 tasks):**   Given multivariate sensor streams, the goal is to flag time points or segments that deviate from regular behavior. Evaluation is by F1-score. Datasets (SMD, MSL, SMAP, SWaT, PSM) come from industrial or telemetry settings, covering diverse systems where faults are rare and heterogeneous.

**Classification (10 tasks):**   Each time series is assigned a single label (e.g., class of gesture, health condition, speaker vowel). The metric is accuracy. The suite spans low to high-dimensional settings (3 to 963 variables) and short to moderate lengths (29–1751 timesteps): EthanolConcentration (alcohol industry), FaceDetection (face signals), Handwriting (motion), Heartbeat (health), JapaneseVowels (voice), PEMS-SF (transportation), SelfRegulationSCP1/2 (health), SpokenArabicDigits (audio), and UWaveGestureLibrary (gestures).

---

[3]For the long-term forecasting category, all of those 9 tasks are sampled into 4 different sequence length, leading to 4 different experiences per tasks (9*4 = 36 tasks).

**Imputation (12 tasks):**   The objective is to reconstruct missing values in partially observed multivariate sequences. Performance is measured by MSE on the masked entries. TSL uses electricity and weather datasets (ETTh1/2, ETTm1/2, Electricity, Weather) with varying dimensions (7–321) and cadences (15 min to hourly for electricity; 10 min for weather).

**Long-term Forecasting (36 tasks):**   Models must predict extended horizons from historical context; TSL evaluates MSE across multiple horizon settings. The category aggregates nine base datasets: ETTh1/2, ETTm1/2, Electricity, Weather, Traffic, Exchange, ILI. Those 9 datasets are sampled at 4 different forecast lengths (96, 192, 336, 720) to reach 36 tasks in total. Series vary widely in dimension (7–862) and domain (energy, environment, transportation, economics, health).

**Short-term Forecasting (6 tasks):**   Univariate forecasting across the six M4 frequencies (Yearly, Quarterly, Monthly, Weekly, Daily, Hourly). TSL reports OWA to compare against classical statistical baselines while normalizing across heterogeneous scales and frequencies. These tasks probe local dynamics, and near-term extrapolation ability.

## D   ALL 10 CONFIGURATIONS OF OUR EXPERIMENTS

The following table shows the 10 configurations (set of Hyper-Parameters) used in our experiments with Time Series Library (TSL). All 69 tasks in TSL were tested with the 10 following configuration.

| Configuration | Layers | Memory Units | Memory Dimension | Model Dimension | Memory Connectivity |
|:---:|:---:|:---:|:---:|:---:|:---:|
| Run 1 | 1 | 4 | 100 | 64 | 0.05 |
| Run 2 | 2 | 4 | 128 | 128 | 0.05 |
| Run 3 | 4 | 8 | 64 | 32 | 0.20 |
| Run 4 | 1 | 2 | 512 | 128 | 0.025 |
| Run 5 | 1 | 4 | 128 | 64 | 0.05 |
| Run 6 | 2 | 8 | 64 | 64 | 0.10 |
| Run 7 | 3 | 4 | 32 | 64 | 0.25 |
| Run 8 | 4 | 2 | 64 | 128 | 0.125 |
| Run 9 | 2 | 16 | 32 | 32 | 0.25 |
| Run 10 | 1 | 16 | 64 | 64 | 0.125 |

Table 2: EST hyperparameter configurations explored in the experimental evaluation.

# E  BEST CONFIGURATION AND SCORE FOR EACH TASK

Table 3: Best model configuration for Anomaly Detection tasks

| Task | layer | mem_units | mem_dim | model_dim | connectivity | F1-Score |
|---|---|---|---|---|---|---|
| anomaly_detection_SMD | 2 | 4 | 128 | 128 | 0.05 | 0.7915 |
| anomaly_detection_PSM | 1 | 16 | 64 | 64 | 0.125 | 0.9305 |
| anomaly_detection_MSL | 1 | 4 | 128 | 64 | 0.05 | 0.8224 |
| anomaly_detection_SMAP | 4 | 2 | 64 | 128 | 0.125 | 0.8294 |
| anomaly_detection_SWAT | 4 | 2 | 64 | 128 | 0.125 | 0.8889 |

Table 4: Best model configuration for Imputation tasks

| Task | layer | mem_units | mem_dim | model_dim | connectivity | MSE |
|---|---|---|---|---|---|---|
| imputation_ETTh1_mask_0.125 | 1 | 4 | 128 | 64 | 0.05 | 0.08205 |
| imputation_ETTh1_mask_0.25 | 1 | 4 | 100 | 64 | 0.05 | 0.1328 |
| imputation_ETTh1_mask_0.375 | 1 | 16 | 64 | 64 | 0.125 | 0.1676 |
| imputation_ETTh1_mask_0.5 | 1 | 2 | 512 | 128 | 0.025 | 0.2120 |
| imputation_ECL_mask_0.125 | 1 | 4 | 128 | 64 | 0.05 | 0.1883 |
| imputation_ECL_mask_0.25 | 1 | 4 | 100 | 64 | 0.05 | 0.1940 |
| imputation_ECL_mask_0.375 | 1 | 4 | 100 | 64 | 0.05 | 0.2001 |
| imputation_ECL_mask_0.5 | 1 | 4 | 100 | 64 | 0.05 | 0.2037 |
| imputation_weather_mask_0.125 | 1 | 4 | 100 | 64 | 0.05 | 0.06920 |
| imputation_weather_mask_0.25 | 1 | 4 | 100 | 64 | 0.05 | 0.04032 |
| imputation_weather_mask_0.375 | 1 | 4 | 128 | 64 | 0.05 | 0.05309 |
| imputation_weather_mask_0.5 | 2 | 16 | 32 | 32 | 0.25 | 0.06432 |

Table 5: Best model configuration for Long-term Forecast tasks

| Task | layer | mem_units | mem_dim | model_dim | connectivity | MSE |
|---|---|---|---|---|---|---|
| long_term_forecast_ili_24 | 2 | 4 | 128 | 128 | 0.05 | 2.668 |
| long_term_forecast_ili_36 | 2 | 4 | 128 | 128 | 0.05 | 2.485 |
| long_term_forecast_ili_48 | 3 | 4 | 32 | 64 | 0.25 | 2.357 |
| long_term_forecast_ili_60 | 3 | 4 | 32 | 64 | 0.25 | 2.703 |
| long_term_forecast_Exchange_96 | 2 | 8 | 64 | 64 | 0.10 | 0.1175 |
| long_term_forecast_Exchange_192 | 3 | 4 | 32 | 64 | 0.25 | 0.2060 |
| long_term_forecast_Exchange_336 | 2 | 8 | 64 | 64 | 0.10 | 0.3603 |
| long_term_forecast_Exchange_720 | 2 | 8 | 64 | 64 | 0.10 | 0.9048 |
| long_term_forecast_traffic_96 | 2 | 4 | 128 | 128 | 0.05 | 0.6540 |
| long_term_forecast_traffic_192 | 2 | 8 | 64 | 64 | 0.10 | 1.191 |
| long_term_forecast_traffic_336 | 2 | 8 | 64 | 64 | 0.10 | 1.33 |
| long_term_forecast_traffic_720 | 3 | 4 | 32 | 64 | 0.25 | 1.392 |
| long_term_forecast_ETTh1_96 | 1 | 2 | 512 | 128 | 0.025 | 0.4625 |
| long_term_forecast_ETTh1_192 | 2 | 8 | 64 | 64 | 0.10 | 0.6754 |
| long_term_forecast_ETTh1_336 | 2 | 16 | 32 | 32 | 0.25 | 0.7411 |
| long_term_forecast_ETTh1_720 | 3 | 4 | 32 | 64 | 0.25 | 0.7425 |
| long_term_forecast_ETTh2_96 | 2 | 4 | 128 | 128 | 0.05 | 0.3475 |
| long_term_forecast_ETTh2_192 | 2 | 8 | 64 | 64 | 0.10 | 0.4360 |
| long_term_forecast_ETTh2_336 | 3 | 4 | 32 | 64 | 0.25 | 0.4673 |
| long_term_forecast_ETTh2_720 | 2 | 16 | 32 | 32 | 0.25 | 0.4784 |
| long_term_forecast_ECL_96 | 2 | 8 | 64 | 64 | 0.10 | 0.4258 |

| Task | layer | mem_units | mem_dim | model_dim | connectivity | MSE |
|------|-------|-----------|---------|-----------|--------------|-----|
| long_term_forecast_ECL_192 | 2 | 8 | 64 | 64 | 0.10 | 0.4753 |
| long_term_forecast_ECL_336 | 2 | 8 | 64 | 64 | 0.10 | 0.6741 |
| long_term_forecast_ECL_720 | 2 | 8 | 64 | 64 | 0.10 | 0.8829 |
| long_term_forecast_weather_96 | 1 | 2 | 512 | 128 | 0.025 | 0.1810 |
| long_term_forecast_weather_192 | 2 | 8 | 64 | 64 | 0.10 | 0.2309 |
| long_term_forecast_weather_336 | 2 | 8 | 64 | 64 | 0.10 | 0.2848 |
| long_term_forecast_weather_720 | 1 | 4 | 128 | 64 | 0.05 | 0.4101 |
| long_term_forecast_ETTm1_96 | 1 | 4 | 100 | 64 | 0.05 | 0.3549 |
| long_term_forecast_ETTm1_192 | 2 | 16 | 32 | 32 | 0.25 | 0.4605 |
| long_term_forecast_ETTm2_96 | 1 | 2 | 512 | 128 | 0.025 | 0.1879 |
| long_term_forecast_ETTm2_192 | 2 | 16 | 32 | 32 | 0.25 | 0.2712 |
| long_term_forecast_ETTm2_336 | 2 | 16 | 32 | 32 | 0.25 | 0.3385 |
| long_term_forecast_ETTm2_720 | 2 | 16 | 32 | 32 | 0.25 | 0.4383 |

Table 6: Best model configuration for Classification tasks

| Task | layer | mem_units | mem_dim | model_dim | connectivity | Accuracy |
|------|-------|-----------|---------|-----------|--------------|----------|
| classification_EthanolConcentration | 1 | 4 | 128 | 64 | 0.05 | 0.3042 |
| classification_FaceDetection | 2 | 16 | 32 | 32 | 0.25 | 0.6722 |
| classification_Handwriting | 1 | 16 | 64 | 64 | 0.125 | 0.4365 |
| classification_Heartbeat | 3 | 4 | 32 | 64 | 0.25 | 0.7707 |
| classification_JapaneseVowels | 1 | 16 | 64 | 64 | 0.125 | 0.9568 |
| classification_PEMS_SF | 4 | 8 | 64 | 32 | 0.2 | 0.8786 |
| classification_SelfRegulationSCP1 | 1 | 2 | 512 | 128 | 0.025 | 0.9181 |
| classification_SelfRegulationSCP2 | 2 | 8 | 64 | 64 | 0.1 | 0.5889 |
| classification_SpokenArabicDigits | 1 | 4 | 128 | 64 | 0.05 | 0.9882 |
| classification_UWaveGestureLibrary | 1 | 16 | 64 | 64 | 0.125 | 0.8938 |

Table 7: Short-term Forecast

| Task | layer | mem_units | mem_dim | model_dim | connectivity | OWA |
|------|-------|-----------|---------|-----------|--------------|-----|
| short_term_forecast_m4_Monthly | 2 | 4 | 128 | 128 | 0.05 | 0.928 |
| short_term_forecast_m4_Yearly | 1 | 16 | 64 | 64 | 0.125 | 0.934 |
| short_term_forecast_m4_Quarterly | 1 | 16 | 64 | 64 | 0.125 | 0.928 |
| short_term_forecast_m4_Weekly | 1 | 16 | 64 | 64 | 0.125 | 0.928 |
| short_term_forecast_m4_Daily | 1 | 16 | 64 | 64 | 0.125 | 0.929 |
| short_term_forecast_m4_Hourly | 2 | 16 | 32 | 32 | 0.25 | 0.928 |

# F    MEMORY FOOTPRINTS

Figure 11 reports the peak GPU memory usage of different models when trained with 1M parameters on an NVIDIA Quadro RTX 5000. We observe that recurrent models such as EST suffer from significantly higher memory requirements (7,035 MB) compared to Transformer-based architectures such as iTransformer (1,251 MB) or TimesNet (817 MB). PatchTST reaches the highest footprint (8,615 MB), while lightweight architectures like Mamba (343 MB) and the vanilla Transformer (611 MB) remain more efficient.

This comparison highlights a major trade-off of the EST design: while its recurrent working memory mechanism provides a static number of operations (FLOPs) per steps, it comes at the expense of memory efficiency during training.

**Peak Memory Footprints for 1M Parameters Models**
(Measured while training on a NVIDIA Quadro RTX 5000)

Figure 11: Peak memory footprints (in MB) during training for models with 1M parameters, evaluated on an NVIDIA Quadro RTX 5000.

