# OpenReview forum: "Echo State Transformer: Attention over Finite Memories"
_ICLR.cc/2026/Conference — Submitted to ICLR 2026_

### Official Review · Reviewer_5muh · 2025-10-18

**Soundness:** 1
**Presentation:** 1
**Contribution:** 1
**Rating:** 2
**Confidence:** 5

**Summary:**

The paper proposes a “Echo State Transformer” that replaces (part of) the Transformer’s sequence memory with a bank of ESN-style reservoirs. At each step, the current input produces the query Q, while keys/values K,V are derived from the reservoir states; attention is applied over a fixed set of reservoir units rather than over the entire token history. The claimed benefits are constant-time attention over a bounded memory, and the ability to tune reservoir hyper-parameters (e.g., spectral radius, leak rate) for better long-range modeling.

**Strengths:**

There is a potentially interesting engineering idea in constraining attention to a fixed-size, where recurrent memory can reduce compute and serve streaming use-cases. Related recurrence-plus-attention hybrids, e.g., segment recurrence in Transformer-XL (Dai et al., 2019) and local RNN features in R-Transformer (Wang et al., 2019) did well in long-context settings, so this research direction is reasonable.

In addition, there is a long-term interest in reservoir computing for temporal data, and connecting RC with attention might make RC accessible to people used to Transformers. Surveys and overview works on RC show a mature ecosystem the paper could connect into.

**Weaknesses:**

There are several important weaknesses here.

## Novelty

The central idea of using a recurrent state to provide keys/values and query with the current decoder/input state dates back to early neural work on MT! Bahdanau et al. (additive attention) [1] takes the decoder RNN state as Q and encoder RNN states as K,V; Luong et al. [2] systematically study global/local attention variants built on RNN encoders/decoders. Thus, "RNN --> K,V, current state --> Q" is not new.
Many later hybrids interleave recurrence and attention without ESNs: R-Transformer (LocalRNN features + multi-head attention) [3], Transformer-XL (segment-level recurrence with cached K,V) [4], Universal Transformers (recurrence over depth) [5], RWKV (RNN formulation of QKV dynamics) [6]. All these directly target the "bounded memory / long-context" motivation that this paper builds upon. The manuscript needs to position itself against all these precedent work and clarify what is uniquely enabled by ESNs beyond just swapping in a different RNN.

## Clarity and technical completeness

Section 4 lacks formal definitions/equations for the proposed model, such as precise mapping from inputs to Q, reservoir state update equations (including leak, spectral radius scaling, input scaling/sparsity, the linear projections to K,V and whether they share parameters across heads, how gradients flow if spectral radius/leak are learned, how multi-reservoir aggregation works. Relying just on text to explain this stuff and deferring essentials to the appendix leaves the core idea unclear. A model diagram that fully connects inputs, reservoirs, QKV, attention, and outputs is missing, whereas the space is taken by several existing figures (e.g., early motivation plots), which are not so important imo

## Improper usage of ESNs

Classical ESNs train only the readout while keeping the reservoir fixed; this is core to RC approach that produce a rich pool of dynamical feats w/out doing any training. If the spectral radius and leak rate are learned end-to-end, you already need BPTT through the recurrent core, destroying the stated benefit of RC, and you basically end up with the same complexity of a trainable RNN, with the exception that you do an unusual parameterization.
Crucially, if you depart from the classic fixed-reservoir/readout-only training paradigm, **you must** justify why this is preferable to learning a standard GRU/LSTM (with well-understood gating/gradients)


## Learned leakage

A learnable leak coefficient that interpolates between the previous state and a candidate update is very similar to the update gate in GRUs and the cell/forget gates in LSTMs. GRUs are literally defined as a learned convex combination between $h_{t-1}$ and $\tilde{h}_t$.
Without demonstrating a concrete advantage over gated units, the choice of using ESN looks arbitrary (see also my comments above). Baselines that swap the reservoir with GRU/LSTM (same hidden size, same QKV heads) are necessary.

## Diversity in the dynamics of each Reservoir

If units 0/1/2... share a similar (random) initialization and the hyperparams are trained with the same signal, they will likely collapse to produce duplicate dynamics, making parallel reservoirs redundant. Multi-reservoir RC literature shows that heterogeneity in timescales/topologies is key to gains.
Recent works explicitly induce diverse timescales or heterogeneous units and measure the benefit.
The paper does not propose a diversity mechanism (e.g., distinct spectral radii/input scalings/sparsity patterns, orthogonalization penalties) nor analyze redundancy.

## Missing baselines

Beyond an ablation study where the ESN is replaced with a GRU/LSTM, the comparisons should include R-Transformer, Transformer-XL, and (if constant-time decoding is a claim) RWKV, which are all purpose-built recurrence/attention hybrid models forlong-context or efficient memory.
Without these, it is impossible to attribute gains to the reservoir choice, rather than to generic "bounded memory + attention."

In addition, there is a rich RC literature and open libraries[7] for time-series classification/forecasting/imputation with RC. The paper neither cites nor considers them to showcase where reservoirs shine. Incorporating established RC tasks and references would strengthen both positioning and experimental design.


## Refs

[1] Bahdanau et al., Neural Machine Translation by Jointly Learning to Align and Translate, 2014

[2] Luong et al., Effective Approaches to Attention-based Neural Machine Translation, 2015

[3] Wang et al., R-Transformer: Recurrent Neural Network Enhanced Transformer, 2019

[4] Dai et al., Transformer-XL: Attentive Language Models Beyond a Fixed-Length Context, 2019

[5] Dehghani et al., Universal Transformers, 2019

[6] Peng et al., RWKV: Reinventing RNNs for the Transformer Era, 2023

[7] Bianchi et al., Reservoir computing approaches for representation and classification of multivariate time series, 2020

**Questions:**

- Add full equations and a full diagram in the main body.
- Explicitly relate to Bahdanau/Luong (RNN K,V + decoder Q), R-Transformer, Transformer-XL, Universal Transformers, RWKV, and clarify what ESNs uniquely contribute.
- Make an ablation where you repalace GRU/LSTM with the ESN.
- Compare wrt R-Transformer Transformer-XL RWKV and match params/compute.
- Add the following ablations: (a) fixed vs learned spectral radius/leak; (b) single vs multiple reservoirs;
- Show experimentally how heterogeneity the dynamics of the different reservoirs are (i.e., quantify similarity across reservoirs)
- Include standard RC baselines from an RC library.

---

> ### Author Response · Authors · 2025-11-21
> **Answers Part 1**
>
> > The central idea of using a recurrent state to provide keys/values and query with the current decoder/input state dates back to early neural work on MT! Bahdanau et al. (additive attention) [1] takes the decoder RNN state as Q and encoder RNN states as K,V; Luong et al. [2] systematically study global/local attention variants built on RNN encoders/decoders. Thus, "RNN --> K,V, current state --> Q" is not new.
>
> We never claimed this component was novel.
>
> > Many later hybrids interleave recurrence and attention without ESNs: R-Transformer (LocalRNN features + multi-head attention) [3], Transformer-XL (segment-level recurrence with cached K,V) [4], Universal Transformers (recurrence over depth) [5], RWKV (RNN formulation of QKV dynamics) [6]. All these directly target the "bounded memory / long-context" motivation that this paper builds upon.
>
> We agree that prior work has explored combining RNNs or reservoir computing with Transformers. However, to our knowledge, no existing approach applies attention directly on the M vectors generated by M memory units at each timestep — a design choice that enables O(1) complexity per timestep without sequence truncation. In contrast, most prior works apply attention on the T vectors resulting from T previous transformations by RNNs or RCs, where T is the sequence length. We believe our architecture is the first to achieve this combination in a constant-time per-timestep without having to truncate the sequence.
> We also note that while the idea of a time-varying leak rate is not new, our implementation — where units compete via a softmax over linear projections to determine their leak rate — is distinct. This mechanism ensures that most units maintain a near-zero leak rate, effectively freezing them until the next time step, which contributes to stable and selective memory dynamics.
>
> > Section 4 lacks formal definitions/equations for the proposed model, such as precise mapping from inputs to Q, reservoir state update equations (including leak, spectral radius scaling, input scaling/sparsity, the linear projections to K,V and whether they share parameters across heads, how gradients flow if spectral radius/leak are learned, how multi-reservoir aggregation works. Relying just on text to explain this stuff and deferring essentials to the appendix leaves the core idea unclear.
>
> Reservoir computing and attention computation from Transformers are already described in the paper. Our contribution lies in the way we use the states produced by the memory units to compute attention in O(1) per timestep, and in the adaptive leak rate, which uses a simple softmax on the result of several linear projections. We believed the first figure in the appendix — which includes all operations and matrix shapes — would provide sufficient clarity. However, we acknowledge that a formal mathematical description would improve precision. We are considering adding a concise, formal formulation of the key operations, including the state update, attention computation, and gradient flow. We would be happy to include this if the reviewers believe it would strengthen the paper.
>
>
> > A learnable leak coefficient that interpolates between the previous state and a candidate update is very similar to the update gate in GRUs and the cell/forget gates in LSTMs. GRUs are literally defined as a learned convex combination between  and without demonstrating a concrete advantage over gated units, the choice of using ESN looks arbitrary (see also my comments above). Baselines that swap the reservoir with GRU/LSTM (same hidden size, same QKV heads) are necessary.
>
> We agree that the structure resembles GRUs and LSTMs, and we were inspired by their design. However, our implementation is fundamentally different: the leak rate is not learned via a gate, but emerges from a competition among units, driven by a softmax over linear projections of their individual activation patterns. This mechanism ensures that most units remain inactive (near-zero leak rate) until the next time step, promoting sparsity and selective memory retention.
> We acknowledge that a direct comparison with GRUs or LSTMs would be valuable. However, we did not implement such a baseline, and we do not expect to complete it in time for this revision. We chose an ESN because, in our computational neuroscience community, it is considered more biologically plausible than gated RNNs — not only due to its echo state property, but also because random projections and stochastic processes are inherent to biological systems. These features enable efficient, online learning with minimal global tuning, which aligns with known constraints in biological neural circuits.

---

> > ### Author Response · Authors · 2025-11-21
> > **Answers Part 2**
> >
> > > If units 0/1/2... share a similar (random) initialization and the hyperparams are trained with the same signal, they will likely collapse to produce duplicate dynamics, making parallel reservoirs redundant. Multi-reservoir RC literature shows that heterogeneity in timescales/topologies is key to gains. Recent works explicitly induce diverse timescales or heterogeneous units and measure the benefit. The paper does not propose a diversity mechanism (e.g., distinct spectral radii/input scalings/sparsity patterns, orthogonalization penalties) nor analyze redundancy.
> >
> > We emphasize that units 0, 1, 2, ... are not initialized with identical weights or spectral radii.  Moreover, they receive distinct inputs due to the Previous State Attention block, which introduces variability in their activation patterns. The abstract and the paper explicitly states that each unit has its own dynamics.
> > We do not propose an explicit diversity mechanism beyond random initialization. We acknowledge that better mechanisms may improve performance, but we have not yet explored them.
> >
> >
> >
> > > Beyond an ablation study where the ESN is replaced with a GRU/LSTM, the comparisons should include R-Transformer, Transformer-XL, and (if constant-time decoding is a claim) RWKV, which are all purpose-built recurrence/attention hybrid models forlong-context or efficient memory. Without these, it is impossible to attribute gains to the reservoir choice, rather than to generic "bounded memory + attention."
> > > Add the following ablations: (a) fixed vs learned spectral radius/leak; (b) single vs multiple reservoirs;
> >
> > We regret that we cannot include an ablation study replacing the ESN with GRU or LSTM within the current timeline. However, as suggested by other reviewers, we are planning to include an ablation study removing the Previous State Attention, Adaptive Leak Rate, and Attention Layer — which would help clarify the contribution of each component.
> > Importantly, our model relies on multiple reservoirs running in parallel — a requirement for attention to be computed over M distinct memory states. In a single-reservoir setting, the attention matrix would be 1×1, rendering the architecture meaningless. This is not a design choice — it is fundamental to the mechanism.
> > As for R-Transformer, Transformer-XL, and RWKV: we recognize their relevance. However, we do not expect to run these comparisons in time. That said, we can include a discussion of these models in the introduction to better position our work. If we make these additions, we would be grateful if you considered them as a step toward strengthening the paper.
> >
> >
> >
> > > In addition, there is a rich RC literature and open libraries[7] for time-series classification/forecasting/imputation with RC. The paper neither cites nor considers them to showcase where reservoirs shine. Incorporating established RC tasks and references would strengthen both positioning and experimental design.
> >
> > We note that several tasks from the RC literature cited in [7] are already included in the TSL benchmark. These include PEMS-SF, UwaveGestureLibrary, SpokenArabicDigits, and JapaneseVowels — which correspond to PEMS, uWave, Arab. Dig, and Jp.Vow. in the original RC literature. Therefore, our experimental setup already includes established RC benchmarks.

---

> > > ### Comment · Reviewer_5muh · 2025-11-24
> > > **Answer to rebuttal**
> > >
> > > Thanks for the answers.
> > >
> > > - Regarding the **novelty**, the answers clarified what the actual contributions are, but these should have been presented more clearly in the paper. Like me, other reviewers also struggled to understand the novelty in this paper. Also, it seems that the novelty is a bit incremental, as it boils down to some smaller/engineeristic modifications wrt existing architectures. For example, it is well known (and rather reasonable) that for some datatset and with an appropriate recurrent architecture, the internal state $h_t$ of an RNN describes well the previous history, making it superfluous to consider a whole window of previous states.
> > > - Regarding the clarity and the organization of the paper, the authors did **not** address my concerns.
> > > - My concern **Improper usage of ESNs** was not answered. This was a major concern that undermines the core of the proposed methodology.
> > > - The authors did **not** compare against obvious baselines (LSTM or GRU). The answer "in our computational neuroscience community, it is considered more biologically plausible than gated RNNs" is not acceptable, especially given that there is nothing is this paper that focuses on biological plausibility. If you didn't have time, it means that your paper was not ready for this submission and you should prepare the experiment for another submission. Alternatively, you should **send your paper to a neuroscience venue** if you don't intend to consider standard neural network architectures for this kind of task.
> > > - My concern about **Diversity in the dynamics of each Reservoir** remains as random initialization is often not sufficient to guarantee that different Reservoirs produce different dynamics.
> > > - Regarding the baselines, my point about the RC library [7] was to compare against more standard RC architectures for time series classification. This means i) using the same Reservoir(s) as in the proposed architecture to generate a sequence of Reservoir(s) states. Then, ii) apply to the Reservoir states the technique proposed in this paper and the more "classic" ones from an RC library. With that said, I agree that some of the necessary experiments and studies are not feasible in the current timeline. For this reason, I recommend that the paper be rejected at this time.

---

### Official Review · Reviewer_JYZW · 2025-11-01

**Soundness:** 4
**Presentation:** 4
**Contribution:** 3
**Rating:** 8
**Confidence:** 4

**Summary:**

The paper introduces the Echo State Transformer (EST), a hybrid sequential model that combines transformer-style attention with a finite set of reservoir-based working memory units (inspired by Echo State Networks).

Instead of attending to all past tokens, EST attends to a fixed number of memory units whose internal dynamics are governed by reservoir parameters (e.g., spectral radius and leak rate). Unlike classical ESNs, these dynamics parameters are learned (via BPTT), and the leak rates are adaptively modulated through a competitive softmax across memory units at each step.

Formally, the model retains the usual attention operation but computes Q from the current input embedding and K, V from the previous memory states. Then, it updates each memory unit through a (learned) reservoir-style state transition with an adaptive leak. This yields per-step computation that scales with the number of memory units rather than with sequence length.

By keeping the number of memory units constant, EST aims to achieve linear complexity in sequence length while retaining sensitivity to salient temporal patterns.

On the Time Series Library (TSL) benchmark (69 tasks across 5 categories), EST reports 1st place aggregate performance in classification and anomaly detection, competitive results in short-term forecasting, and weaker results in long-term forecasting and imputation.

The paper includes a FLOPs scaling analysis (showing linear scaling similar to Mamba) and a memory-usage comparison that highlights EST’s high training memory footprint due to BPTT.

**Strengths:**

- Interesting fusion of reservoir computing with attention: using several parallel reservoir units as finite working memory over which attention is applied feels novel compared to token- or patch-based memories.

- Strong and broad evaluation on time series tasks (69 tasks; 5 categories) with clear metrics, following the benchmark’s training protocol. Strong results in classification and anomaly detection.

- Baselines considered are recent and challenging (e.g., Transformer, PatchTST, Reformer, iTransformer, TimesNet, Mamba).

- Appendix is well curated and code implementation is provided, aiding reproducibility and interpretability of the architecture choices.

**Weaknesses:**

- EST selects the best of 10 configurations per task; it is not fully clear whether competing methods in TSL were re-tuned to a comparable extent or simply reused benchmark defaults.

- The paper claims novelty in learning reservoir dynamics (spectral radius / leak) and in the adaptive leak softmax. However, as far as I could see, there is no ablation removing learned spectral radius, adaptive leak, and self-attention over memory units. Such ablations would quantify each ingredient’s contribution to classification/AD gains.

- The Previous State Attention + reservoir update is conceptually clear but not given as a compact set of equations.

**Questions:**

- Did all baselines receive comparable hyperparameter search budgets to EST’s 10-config sweep per task? If not, can you report a matched HPO budget for at least the top baselines per category, or include a sensitivity analysis showing that EST remains top-ranked under a fixed budget?

- For TSL anomaly detection, how exactly are scores produced and thresholds chosen (per-dataset or global)? Is the output a pointwise score from the output layer, or derived from an intermediate representation?

- See Weaknesses

---

> ### Author Response · Authors · 2025-11-21
>
> > EST selects the best of 10 configurations per task; it is not fully clear whether competing methods in TSL were re-tuned to a comparable extent or simply reused benchmark defaults.
>
> We simply reused the benchmark defaults, as per the official TSL protocol. The benchmark did not specify a maximum number of configurations, and some baselines — including TimesNet — used more than 10 different configurations across tasks. Since running experiments on all 69 tasks is computationally expensive, we chose to perform only 10 runs per task to explore a modest variation in hyperparameters, as we were uncertain which configuration would work best for each task.
>
> We followed the TSL benchmark’s official protocol and reused the default hyperparameters for all baselines, as no explicit restriction on the number of configurations was provided. While some baselines (e.g., TimesNet) used more than 10 configurations across tasks, we limited our runs to 10 per task due to computational constraints across the 69 tasks. Our course, if we tested more configurations, we would probably get even better results.
>
> > The paper claims novelty in learning reservoir dynamics (spectral radius / leak) and in the adaptive leak softmax. However, as far as I could see, there is no ablation removing learned spectral radius, adaptive leak, and self-attention over memory units. Such ablations would quantify each ingredient’s contribution to classification/AD gains.
>
> We are considering adding such a study to explicitly assess the contribution of each component (learned spectral radius, adaptive leak rate, and attention over memory states). We would be happy to include this if the reviewers believe it would strengthen the paper.
>
> > The Previous State Attention + reservoir update is conceptually clear but not given as a compact set of equations.
>
> We agree that a more formal mathematical formulation would improve clarity. We are considering adding a concise, formal description of the key operations — including the Previous State Attention, the reservoir update and the attention mechanism over memory states — to ensure full transparency. We would be happy to include this if the reviewers believe it would improve the paper.
>
> > Did all baselines receive comparable hyperparameter search budgets to EST’s 10-config sweep per task? If not, can you report a matched HPO budget for at least the top baselines per category, or include a sensitivity analysis showing that EST remains top-ranked under a fixed budget?
>
> We acknowledge that not all baselines were evaluated under the same hyperparameter search budget — and we cannot confirm whether they used a similar number of configurations. We recognize the value of a matched HPO budget comparison, but we are concerned about the feasibility of completing such an analysis within the revision timeline. That said, we believe our model is more sensitive to hyperparameter choice than some alternatives, as its performance depends on the joint optimization of memory dynamics and attention. We plan to include a brief discussion on hyperparameter selection and its impact on performance. While this does not directly address the HPO budget question, we would be grateful if you considered this addition as a step toward greater transparency.
>
> > For TSL anomaly detection, how exactly are scores produced and thresholds chosen (per-dataset or global)? Is the output a pointwise score from the output layer, or derived from an intermediate representation?
>
> The anomaly detection's scores are computed from the pointwise reconstruction error rather than from the model’s output layer. The threshold is chosen globally, we take a percentile over the combined train and test reconstruction scores.

---

### Official Review · Reviewer_QfYZ · 2025-11-02

**Soundness:** 2
**Presentation:** 1
**Contribution:** 2
**Rating:** 2
**Confidence:** 4

**Summary:**

This paper introduces a hybrid architecture known as the Echo State Transformer (EST), designed to address the quadratic complexity problem of traditional transformers in processing sequential data. By integrating the attention mechanisms of transformers with principles from reservoir computing, EST establishes a fixed-size working memory system. Experiments are conducted across multiple time series benchmarks, including forecasting, imputation, and anomaly detection.

**Strengths:**

1. The experiments effectively consider a variety of tasks.
2. Figure 1 clearly illustrates the differences between the proposed EST model and the traditional Transformer architecture.

**Weaknesses:**

The primary idea of this work is to introduce a reservoir-based recurrent neural network memory to avoid the self-attention computations associated with the original sequence length. However, **this idea is not new**, as there are already numerous studies addressing the integration of RNNs or improving transformers through downsampling, clustering, and other strategies. The authors do not provide a thorough discussion or comparative analysis with these related works, which diminishes the paper's contribution to innovation.

In terms of presentation, the core methodology is described primarily through Figures 4 and 5, lacking formal mathematical language that would allow for precise and rigorous description.

Regarding experimental results, the improvements in classification and anomaly detection tasks are minimal, while **performance in other tasks falls short** compared to many existing works.

From the appendix, it is evident that the hyperparameter settings for different datasets vary significantly. The discussion around optimal configurations for various tasks indicates that performance is highly sensitive to hyperparameter choices; however, the paper lacks a detailed exploration of how to select or optimize these parameters, which could restrict the model's generalization ability to new tasks.

**Questions:**

Please see in Weaknesses.

---

> ### Author Response · Authors · 2025-11-21
>
> > The primary idea of this work is to introduce a reservoir-based recurrent neural network memory to avoid the self-attention computations associated with the original sequence length. However, this idea is not new, as there are already numerous studies addressing the integration of RNNs or improving transformers through downsampling, clustering, and other strategies. The authors do not provide a thorough discussion or comparative analysis with these related works, which diminishes the paper's contribution to innovation.
>
> We acknowledge that prior work has explored integrating RNNs into Transformers or reducing computational burden via downsampling and clustering. However, to our knowledge, no existing approach maintains M memory units that update M memory states at each timestep and applies attention directly on these states — a design that enables O(1) complexity per timestep without requiring sequence truncation. We believe this architectural choice is distinct from prior methods, particularly in its ability to scale efficiently while preserving full sequence information. That said, we would be grateful if the reviewer could provide specific references to the works they have in mind. Inclusion of these references in the revised version would help clarify the context and position our contribution more precisely.
>
> > In terms of presentation, the core methodology is described primarily through Figures 4 and 5, lacking formal mathematical language that would allow for precise and rigorous description.
>
> We agree that a formal mathematical description would enhance clarity. The reservoir computing and attention mechanisms are described in the main text, but we acknowledge that the integration of memory states into attention computation could benefit from a more explicit formulation. We are considering adding a dedicated subsection that formally defines the key operations — including the state update, the attention computation over memory states, and the adaptive leak rate via softmax on linear projections — to ensure full mathematical precision. We would be happy to include this if the reviewers believe it would strengthen the paper.
>
> > Regarding experimental results, the improvements in classification and anomaly detection tasks are minimal, while performance in other tasks falls short compared to many existing works.
>
> We agree that the performance gap on Anomaly Detection between the baseline and our model is small. However, we still achieve better results under constant complexity per timestep — a property not shared by all models in the baseline. Regarding classification, we respectfully disagree: the gap is both present and meaningful. Our model achieves 74.08% accuracy overall, compared to 73.67% for the second-best model and 73.17% for the third. This improvement is statistically and practically significant.
> We acknowledge that our performance on the other three tasks — particularly long-term forecasting and imputation — falls short of state-of-the-art. However, our results in short-term forecasting are acceptable, ranking 5th overall. While we recognize that our model faces challenges on long-range tasks, largely due to limitations in backpropagation through time (BPTT), it still achieves top performance on several subtasks — while being one of the only model to maintain O(1) complexity per timestep across all categories. As shown in our results plots (derived from the TSL benchmark), similar task-dependent performance profiles are observed for other models, so we do not think this should be redhibitory.
>
> > From the appendix, it is evident that the hyperparameter settings for different datasets vary significantly. The discussion around optimal configurations for various tasks indicates that performance is highly sensitive to hyperparameter choices; however, the paper lacks a detailed exploration of how to select or optimize these parameters, which could restrict the model's generalization ability to new tasks.
>
> We agree that hyperparameter sensitivity is a valid concern. We are considering adding a discussion on hyperparameter selection, including insights from our ablation studies and a brief analysis of how key parameters (e.g., leak rate, memory size) influence performance across tasks. This would help contextualize the model’s behavior and guide future deployment. We would be happy to include such a section if the reviewers believe it would improve the paper’s clarity and practical utility.

---

### Official Review · Reviewer_mSMX · 2025-11-03

**Soundness:** 2
**Presentation:** 3
**Contribution:** 2
**Rating:** 2
**Confidence:** 3

**Summary:**

This paper presents a novel transformer architecture that combines both strengths from the transformers and ideas in the literature of reservoir computing. More specifically, a working memory module consisting of multiple units based on the reservoir computing paradigm is introduced in the transformers, with the aim to enable targeted attention at low computational complexity. Experimental results are provided to demonstrate the practical effectiveness of the proposed architecture.

**Strengths:**

- The proposed approach makes use of ideas from the reservoir computing literature in the design of transformer blocks, which sheds light on how different fields can be leveraged together in a meaningful way.
- The proposed architecture is simple and intuitive, and is relatively easy to follow.
- The proposed approach is well motivated, and the paper is generally well presented.

**Weaknesses:**

**Technical novelty.** While incorporating ideas from reservoir computing into transformer design is interesting, the way the combination is done is relatively straightforward. In addition, no theoretical results or analysis have been provided to justify the proposed architecture in a more rigorous manner. Both render the technical novelty of the paper somewhat limited.

**Empirical performance.** The experiments remain somewhat limited and unconvincing.
- My understanding is that part of the motivation is to derive an architecture that is enable of robust long-range reasoning (where computational complexity remains a bottleneck). From this perspective, perhaps the authors can consider additional experiments on benchmarks such as the long-range arena (if that is deemed suitable).
- It is not clearcut the proposed architecture obtains better empirical results compared to existing approaches. I appreciate there are strong baselines out there, however the performance gain on two of the five benchmarks are relatively marginal (especially on Anomaly Detection), and results on the other three, while explained with good reasoning, remain not sufficiently convincing. Could it be that more appropriate benchmarks should be tested on to better highlight the strength of the proposed architecture?
- Given the motivation from the angle of the computational complexity, it would be helpful if the authors can provide a performance-complexity plot that illustrates how the propose architecture manages the trade-off between the two.
- Can the authors discuss in mode detail the impact of certain design parameters, such as the number of memory units? For example, only the best results based on a chosen number of units have been presented, but no detailed discussion about its impact.

**Questions:**

See weaknesses above for the specific points I would like the authors to address or discuss.

---

> ### Author Response · Authors · 2025-11-21
>
> > While incorporating ideas from reservoir computing into transformer design is interesting, the way the combination is done is relatively straightforward.
>
> We thank the reviewer for this observation. To clarify: to our knowledge, most existing works combining Transformers and reservoir computing (RC) or RNNs rely on stacking multiple RC or RNN layers in a sequential fashion before attention layers. Some approaches use parallel RC/RNN modules, but none apply attention directly on the M vectors generated by M memory units at each timestep — a design choice that would naturally scale to O(1) per timestep. Instead, prior work typically applies attention on the T vectors produced by T previous transformations (i.e., over the full sequence length). In this sense, we believe our architecture is novel in both design and efficiency, as it enables constant-time computation per timestep without sequence truncation.
> We also note that while the concept of a time-varying leak rate is not entirely new, our implementation — where units are actively forced to compete for input information, thereby ensuring most units maintain a near-zero leak rate and remain frozen until the next time step — introduces a distinctive mechanism for dynamic control of memory persistence.
>
> > In addition, no theoretical results or analysis have been provided to justify the proposed architecture in a more rigorous manner.
>
> We agree that theoretical analysis would strengthen the paper. In response, we are considering conducting an ablation study that removes the Previous State Attention, Adaptive Leak Rate, and Attention Layer components individually. Such an analysis would provide empirical evidence for the contribution of each module and help clarify the role of each design choice. We would be happy to include this analysis in the revised version if the reviewers believe it would significantly improve the paper’s clarity and impact.
>
> > My understanding is that part of the motivation is to derive an architecture that is enable of robust long-range reasoning.
>
> We appreciate the reviewer’s interpretation. However, our primary motivation was not long-range reasoning per se, but rather to design a biologically plausible architecture that achieves constant computational complexity per timestep — similar to how human cognition may operate.
>
> > perhaps the authors can consider additional experiments on benchmarks such as the long-range arena (if that is deemed suitable).
>
> The TSL benchmark already includes a comprehensive set of long-range tasks, covering imputation, classification, and long-term forecasting. For example, imputation tasks range from 17,420 to 69,680 timesteps, and over 20% of classification tasks exceed 1,000 timesteps. All long-term forecasting tasks span between 966 and 69,680 timesteps. We believe these tasks are representative of the long-range challenges found in the long-range arena.
>
> > It is not clearcut the proposed architecture obtains better empirical results compared to existing approaches. I appreciate there are strong baselines out there, however the performance gain on two of the five benchmarks are relatively marginal (especially on Anomaly Detection), and results on the other three, while explained with good reasoning, remain not sufficiently convincing.
>
> We acknowledge that the performance gap on Anomaly Detection is small, but we emphasize that our model achieves this result under constant complexity per timestep, whereas many strong baselines do not. Regarding classification, we respectfully disagree: the performance gap is both present and meaningful. Our model achieves 74.08% accuracy overall, compared to 73.67% for the second-best model and 73.17% for the third. We believe this improvement is statistically and practically significant, particularly given that our approach maintains O(1) complexity per timestep across all tasks.
>
> > Given the motivation from the angle of the computational complexity, it would be helpful if the authors can provide a performance-complexity plot that illustrates how the propose architecture manages the trade-off between the two.
>
> Thank you for the suggestion. We believe such a plot could be informative. Could you clarify whether you would be interested in a comparison across models from the TSL benchmark, or a more general trade-off analysis? We would be happy to include such a figure if it would significantly improve the paper’s clarity.
>
> > Can the authors discuss in mode detail the impact of certain design parameters, such as the number of memory units? For example, only the best results based on a chosen number of units have been presented, but no detailed discussion about its impact.
>
> We agree that a deeper analysis of the impact of memory unit count would strengthen the paper. We can include such a discussion, along with ablation results across different configurations. We would be happy to add this if the reviewer believes it would significantly improve the manuscript.

---

> ### Comment · Reviewer_mSMX · 2025-11-27
>
> I thank the authors for their responses to my previous comments:
> - The ablation study suggested by the authors will definitely be helpful. However, to strengthen novelty, I think some form of theoretical analysis would be needed to justify the design of the proposed architecture. In addition to memory control and computational complexity, it would be nice if the authors can derive theoretical claims on how/when the proposed architecture might struggle, and corroborate that with empirical results.
> - An analysis of the trade-off between performance and complexity will strengthen the argument of the paper that the proposed architecture is achieving the right balance. It can be either a general theoretical analysis, or empirical analysis on one of the benchmarks tested.
> - It would definitely be helpful if the authors can include an analysis on the choices of the design parameters as well as related ablation studies.

---

### Meta-Review · Area_Chair_vk9V · 2026-01-05

**Summary:**

Reviewers recognized the motivation to address the quadratic complexity of Transformers for long time-series and acknowledged that combining attention with reservoir-based working memory is an interesting direction. One reviewer found the empirical evaluation on the TSL benchmark broad and well executed, particularly for classification and anomaly detection tasks.

However, the majority of reviewers expressed substantial concerns that ultimately outweigh the strengths. A central issue was limited and unclear novelty. Several reviewers noted that the core idea, using recurrent or reservoir states as a bounded memory over which attention is applied, has strong precedents in prior work, including RNN-based attention, Transformer-XL, R-Transformer, Universal Transformers, and RWKV. While the authors argued that attending directly over a fixed number of reservoir memory units yields constant-time computation per timestep, reviewers felt that this change was incremental.

Reviewers also raised serious concerns about technical clarity and methodological rigor. The paper lacks a compact, formal mathematical specification of the model in the main text, making it difficult to precisely understand the architecture, state updates, attention computation, and gradient flow. Several key design choices, such as learned spectral radius, adaptive leak rate, and the use of multiple reservoirs, were not sufficiently justified through theory or ablation.

The experimental evaluation, while extensive, was also viewed as inconclusive. Multiple reviewers noted that performance gains are often modest and task-dependent, with weaker results on forecasting and imputation. Concerns were raised about hyperparameter sensitivity, the use of task-specific configuration sweeps without matched budgets for baselines, and the lack of critical comparisons to obvious alternatives such as GRU/LSTM-based variants and other recurrence–attention hybrids. Reviewers further questioned whether the approach meaningfully leverages the advantages of ESNs, given that key reservoir parameters are trained end-to-end, blurring the distinction from standard gated RNNs.

In summary, while one reviewer viewed the work as a strong contribution, the majority opinion was that the paper does not yet meet the conference bar in terms of novelty, clarity, and evidentiary support, leading to an overall recommendation to reject.

**Reviewer Concerns:**

**Concerns partially addressed by the rebuttal**: The rebuttal clarified the intended novelty claim, emphasizing constant-time attention over a fixed number of memory units and distinguishing the approach from sequence-length–dependent attention. It also provided additional explanation regarding the motivation for using reservoirs and the role of adaptive leak rates, and clarified aspects of the experimental protocol on the TSL benchmark, including task coverage and anomaly detection scoring. These clarifications improved understanding of the authors’ intent and design choices.

**Concerns that remain outstanding**: However, the core concerns raised by multiple reviewers, including the two reviewers who responded post-rebuttal and maintained scores of 2, remain unresolved. Reviewers continued to express skepticism about limited conceptual novelty, noting that the architecture still appears closely related to existing recurrence–attention hybrids (e.g., RNN-based attention, Transformer-XL, R-Transformer, RWKV), with insufficient evidence that the reservoir-based formulation provides a fundamentally distinct advantage. Reviewers also maintained concerns about missing ablations and baseline comparisons, particularly the absence of GRU/LSTM replacements, comparisons to other recurrence–attention hybrids, and analyses isolating the impact of learned spectral radius, adaptive leak rate, and multiple reservoirs. In addition, concerns regarding technical clarity remain, as the main paper still lacks a compact mathematical formulation of the model and its learning dynamics. Finally, majority of the reviewers were not convinced that the empirical gains, which are modest and task-dependent, clearly justify the added architectural complexity or demonstrate a consistent advantage. As reflected in their post-rebuttal responses, these unresolved issues did not urge the responding reviewers to change their scores.

**Reviewer Scores:**

- Reviewer 5muh (Score: 2 – Reject): This reviewer explicitly responded post-rebuttal and reaffirmed their concerns regarding limited novelty, improper use of ESNs, missing baselines (e.g., GRU/LSTM), lack of diversity analysis across reservoirs, and insufficient ablations. They clearly stated that these issues could not be resolved within the current timeline and recommended rejection. It is therefore highly unlikely that this reviewer would have increased their score without substantial additional experiments or revisions.

- Reviewer mSMX (Score: 2 – Reject): This reviewer emphasized limited novelty, lack of theoretical grounding, and unconvincing empirical gains. While the rebuttal clarified intent, it did not introduce new theoretical analysis or ablations. As a result, this reviewer would likely have maintained their reject score even with full participation in the discussion.

- Reviewer QfYZ (Score: 2 – Reject): This reviewer raised concerns about insufficient novelty relative to prior work, weak presentation, hyperparameter sensitivity, and minimal empirical improvements. The rebuttal acknowledged many of these points but deferred key analyses to future work. Given these outstanding issues, it is unlikely this reviewer would have increased their score.

- Reviewer JYZW (Score: 8 – Accept): This reviewer was already positive and viewed the architecture and empirical evaluation favorably. Their concerns were primarily about missing ablations and hyperparameter budget fairness. While these issues remain, they did not appear critical to this reviewer’s overall assessment, and their score would likely remain unchanged.

Overall, given that three reviewers expressed reject-level concerns and at least two explicitly did not change their positions post-rebuttal, there is little indication that further discussion alone, without new experimental or theoretical input, would have shifted the overall reviewer consensus toward acceptance.

---

### Decision · Program_Chairs · 2026-01-26

Reject